# Rapid decline of neutralizing antibodies against SARS-CoV-2 among infected healthcare workers

Stéphane Marot[1,6✉], Isabelle Malet [1,6], Valentin Leducq[1], Karen Zafilaza[1], Delphine Sterlin[2], Delphine Planas[3,4], Adélie Gothland[1], Aude Jary [1], Karim Dorgham [2], Timothée Bruel [3,4], the Sorbonne Université SARS-CoV-2 Neutralizing Antibodies Study Group*, Sonia Burrel [1], David Boutolleau[1], Olivier Schwartz[3,4], Guy Gorochov[2], Vincent Calvez[1] & Anne-Geneviève Marcelin [1]

There are only few data concerning persistence of neutralizing antibodies (NAbs) among SARS-CoV-2-infected healthcare workers (HCW). These individuals are particularly exposed to SARS-CoV-2 infection and at potential risk of reinfection. We followed 26 HCW with mild COVID-19 three weeks (D21), two months (M2) and three months (M3) after the onset of symptoms. All the HCW had anti-receptor binding domain (RBD) IgA at D21, decreasing to 38.5% at M3 ($p < 0.0001$). Concomitantly a significant decrease in NAb titers was observed between D21 and M2 ($p = 0.03$) and between D21 and M3 ($p < 0.0001$). Here, we report that SARS-CoV-2 can elicit a NAb response correlated with anti-RBD antibody levels. However, this neutralizing activity declines, and may even be lost, in association with a decrease in systemic IgA antibody levels, from two months after disease onset. This short-lasting humoral protection supports strong recommendations to maintain infection prevention and control measures in HCW, and suggests that periodic boosts of SARS-CoV-2 vaccination may be required.

[1] Sorbonne Université, INSERM, Institut Pierre Louis d'Epidémiologie et de Santé Publique (iPLESP), Assistance Publique-Hôpitaux de Paris (AP-HP), Pitié Salpêtrière Hospital, Department of Virology, Paris, France. [2] Sorbonne Université, INSERM, Centre d'Immunologie et des Maladies Infectieuses (CIMI-Paris), AP-HP, Pitié Salpêtrière Hospital, Department of Immunology, Paris, France. [3] Virus and Immunity Unit, Department of Virology, Institut Pasteur, CNRS UMR 3569 Paris, France. [4] Vaccine Research Institute, Creteil, France. [6]These authors contributed equally: Stéphane Marot, Isabelle Malet. *A list of authors and their affiliations appears at the end of the paper. ✉email: stephanesylvain.marot@aphp.fr

A novel human coronavirus, severe acute respiratory syndrome coronavirus 2 (SARS-CoV-2), the causal agent of coronavirus disease 19 (COVID-19), emerged in late 2019 and rapidly spread worldwide, causing a global pandemic[1]. SARS-CoV-2 infection results in a wide range of clinical manifestations, from asymptomatic or mild forms to severe respiratory failure[2]. Healthcare workers (HCW) are particularly exposed to SARS-CoV-2 infection during the management of patients with COVID-19, and several recommendations have been made to protect them[3–5]. Moreover, infected HCW could serve as a source of hospital-acquired COVID-19. It is therefore important to assess the durability of immune protection after SARS-CoV-2 infection in this population.

Serological diagnosis is becoming increasingly important in attempts to understand the extent to which COVID-19 is spreading in the community and to identify individuals who are immunized and potentially "protected" against re-infection. Most serological studies to date have focused on hospitalized patients, and were therefore unable to evaluate the serologic responses in individuals with mild or subclinical infection, including HCW in particular[6–10]. The few studies focusing on this particular population have investigated only seroconversion rates, without considering the persistence of the SARS-CoV-2 antibodies induced over time[11–13]. Recent studies have shown that anti-SARS-CoV-2 antibody levels decrease from 8 weeks after the onset of symptoms and that serum samples from 33% of SARS-CoV-2-infected patients display no neutralizing activity, even 39 days after symptom onset[10,14]. Moreover, the protection correlates of immunity to SARS-CoV-2 infection are not well defined, and little is known about the persistence of antibodies over time. There is some evidence that immune protection against other human coronaviruses (HCoV) is not long-lasting[15], but there is also evidence that antibodies against the severe acute respiratory syndrome coronavirus (SARS-CoV)[16,17] persist for years and display potent neutralizing activity even in the absence of detectable serum immunoglobulin G (IgG) directed against SARS-CoV nucleocapsid protein (N)[18]. Neutralizing antibodies (NAbs) prevent viral infection mostly by blocking the early step of infection, viral entry, especially in interfering with virions binding to their receptor. NAbs can be evaluated in vitro in neutralization tests, and their presence is often correlated with protective immunity[19].

There is now a need to correlate the detection of different isotypes of antibodies produced in response to SARS-CoV-2 infection with their neutralizing activity, to assess the development of protective antibody responses and to study their duration.

Here, we aimed at studying the longitudinal profile of IgG, immunoglobulin A (IgA) and M (IgM) and the neutralizing activity of sera against SARS-CoV-2, for 26 HCW recovering from mild forms of reverse transcription PCR (RT-PCR)-confirmed SARS-CoV-2 infection.

In this work we show that all the studied HCW displayed seroconversion and a NAb response against SARS-CoV-2 from 21 days after symptom onset, which was correlated with the levels of anti-RBD antibodies. However, this neutralizing activity appears to be transient with a decline, or even a loss, of the NAb titers associated with a decrease in systemic IgA antibody levels from 2 months after disease onset.

## Results

**Patient characteristics and serum collection.** From 28 January to 21 March 2020, 26 HCW from Pitié-Salpêtrière University Hospital were enrolled in this study. Their characteristics are presented in Table 1. The median [interquartile (IQR)] age of the

**Table 1 Characteristics of the healthcare workers (HCW) with SARS-CoV-2 infection.**

| Characteristics | Results (n = 26) |
|---|---|
| Male, n (%) | 11 (39.3) |
| Age (years), median [IQR] | 46 [38–53] |
| Occupation, n (%) | |
| Physician | 18 (69.2) |
| Paramedical staff | 5 (19.2) |
| Administrative staff | 3 (11.5) |
| SARS-CoV-2 exposure, n (%) | |
| Professional exposure without COVID-19 patient care activities | 12 (46.2) |
| Involved in COVID-19 patient care activities | 14 (53.8) |
| Including care activity with a high risk of viral aerosolization | 6 (23.1) |
| Time between onset of symptoms and positive RT-PCR result (days), median [IQR] | 2 [1–3] |
| RT-PCR cycle threshold at the time of diagnosis (arbitrary units), median [IQR] | 22.2 [18.7–24.2] |
| Reported symptoms, n (%) | |
| Asthenia | 21 (80.8) |
| Anosmia | 18 (69.2) |
| Headache | 17 (65.4) |
| Ageusia | 15 (57.7) |
| Fever | 15 (57.7) |
| Dry cough | 15 (57.7) |
| Myalgia | 15 (57.7) |
| Shivers | 10 (38.5) |
| Dyspnea | 7 (26.9) |
| Sweats | 7 (26.9) |
| Rhinitis | 6 (23.1) |
| Diarrhea | 5 (19.2) |
| Dizziness | 4 (15.4) |
| Pharyngitis | 3 (11.5) |
| Nausea | 2 (7.7) |
| Chest pain | 2 (7.7) |
| Palpitations | 1 (3.8) |
| Abdominal pain | 1 (3.8) |
| Conjunctivitis | 1 (3.8) |
| Duration of symptoms, n (%) | |
| 1 week | 8 (30.8) |
| 2 weeks | 7 (26.9) |
| 3 weeks | 6 (23.1) |
| 4 weeks | 3 (11.5) |
| >1 month | 2 (7.7) |
| New symptoms after a period of recovery, n (%) | |
| Yes | 5 (19.2) |
| No | 21 (80.8) |

The underlying detailed information is provided additionally within the source data file.

participants was 46 [38–53] years, 11 (42.3%) were men, and most (69.2%) were physicians. The hypothetical exposure to SARS-CoV-2 is thought to have occurred during the management of COVID-19 patients in 14 (53.8%) HCW, 6 (23.1%) of whom performed high-risk aerosol-generating procedures (nasopharyngeal, tracheal and pulmonary sampling, endotracheal intubation, use of high-flow nasal cannula, oxygen masks or non-invasive ventilation). However, 12 HCW (46.2%) were exposed to the virus during professional activities unrelated to patient care.

All the HCW had symptoms consistent with COVID-19 (Table 1). The median duration of symptoms was 2 [1–3] weeks and 5 (19.2%) HCW developed new symptoms after a period of recovery. The new symptoms were principally dysosmia and transient headache, developing a median of 3.5 [2–4] months after the COVID-19 episode.

The median time between the onset of symptoms and positive RT-PCR results was 2 [1–3] days, and the median time from onset of symptoms to blood sampling was 17 [14–23.25] days for the D21 time point, 59 [57–61] days for M2, and 97 [91–100] for M3.

**IgG, IgA, and IgM profile**. At the first time point (D21), all HCW had detectable IgG antibodies against SARS-CoV-2 N protein, IgA antibodies against SARS-CoV-2 receptor-binding domain (RBD), and IgA antibodies against SARS-CoV-2 spike protein (S). In addition, 92.3% (24/26) had detectable anti-RBD IgG antibodies and 80.8% (21/26) had detectable anti-S IgG antibodies. Only 42.3% (11/26) had detectable anti-N/anti-S IgM antibodies. At M2, all HCW remained positive for anti-N IgG antibodies. The rates of seropositivity for anti-S IgG, anti-RBD IgG, and anti-N/anti-S IgM antibodies were 94.1% (16/17), 100% (17/17), and 64.7% (11/17), respectively. Meanwhile, the anti-S IgA seropositivity rate declined to 88.2% (15/17) and that for anti-RBD IgA fell to 47% (8/17). At M3, the seropositivity rate for anti-N IgG had decreased slightly, to 96.2% (25/26). The proportion of anti-S IgG antibodies was 96.2% (25/26), and the proportion of anti-S IgA antibodies continue to decline, reaching 80.8% by this time point (21/26). In parallel, the seropositivity rate for anti-RBD IgG antibodies declined to 92.3% (24/26) and that for anti-RBD IgA fell to 38.5% (10/26). We were able to detect anti-N/anti-S IgM antibodies in 15/26 HCW (57.7%).

We then compared the changes in the levels of these antibodies over time, to assess the intensity of the immune response to SARS-CoV-2 (Fig. 1). Anti-S and anti-RBD IgA antibody levels decreased significantly between D21 and M2 ($p = 0.007$ and $p < 0.001$, respectively) and between D21 and M3 ($p < 0.001$ and $p < 0.001$, respectively). Anti-S and anti-RBD IgG antibody levels did not change significantly between D21 and M3, despite a slight increase in antibody levels at M2 ($p < 0.001$ and $p = 0.04$). Anti-N IgG levels were stable at M2, but decreased significantly between D21 and M3 ($p < 0.001$). Finally, no change in anti-N/anti-S IgM levels were observed between D21 and M2 or M3 ($p = 0.18$ and $p = 0.53$, respectively). Ten serum samples from patients with other respiratory infections caused by HCoV (2 HCoV-HKU1, 3 HCoV-OC43, 3 HCoV-NL63, and 2 HCoV-229E), obtained between 11 and 54 days after RT-PCR diagnosis, were also tested in each immunoassay, to evaluate the specificity of these tests. None of these samples gave positive results with the SARS-CoV-2-specific antibody tests used, demonstrating an absence of cross-reactivity.

**Changes in Nab titers over time**. We analyzed the neutralizing activity of serum samples from the 26 HCW in a virus neutralization test (VNT). At D21, all sera (100%) harbored NAbs with a neutralizing titer ≥1:5, the median neutralizing titer being 1:20 [1:10–1:40] (Fig. 2). At M2, serum samples from 88.2% (15/17) of the HCW were still neutralizing, with a median neutralizing titer of 1:20 [1:5–1:20]. By M3, serum samples from another two HCW had lost their neutralizing activity; serum samples from 84.6% (22/26) of the HCW remained neutralizing, with a median titer of 1:10 [1:5–1:20]. NAb titers decreased significantly between D21 and M2 ($p = 0.03$) and between D21 and M3 ($p < 0.001$). The four HCW whose sera lost their neutralizing capacity had no detectable anti-RBD IgA antibodies at M3, whereas such antibodies were detected when their sera were neutralizing at D21.

We also tested the same 10 HCoV serum samples (2 HCoV-HKU1, 3 HCoV-OC43, 3 HCoV-NL63, and 2 HCoV-229E) to assess cross-neutralization capacities between HCoV and SARS-CoV-2.

None of these serum samples displayed SARS-CoV-2-neutralizing capacity.

**Association of NAb titer with IgG, IgA, and IgM antibody profile**. We first evaluated the relative contributions of each antibody isotype and/or epitopes to the neutralizing capacity of the sera at the various time points. We found that the antibodies most strongly associated with NAb titers were anti-RBD IgG and IgA antibodies (Spearman's rank correlation coefficient ($r_s$): 0.73 [confidence interval 95%, CI 95%: 0.58–0.82]; $p < 0.0001$ and $r_s$: 0.64 [CI 95%: 0.47–0.77]; $p < 0.0001$, respectively). We purified IgA and IgG antibody fractions from three participants at D21 and M3 to evaluate their relative abilities to neutralize SARS-CoV-2 in a neutralization experiment, with syncytium formation in S-Fuse cells as the readout, evaluating the half-maximal neutralizing effective concentration ($EC_{50}$)[20]. We found that the purified IgA had a greater neutralizing capacity than IgG in each of the paired serum samples at the early time point (D21; IgA $EC_{50} <$ IgG $EC_{50}$), whereas the opposite pattern was observed at M3 (Fig. 3).

**Discussion**

We describe here the changes over time in IgG, IgA, and IgM antibody profiles associated with the neutralizing activity of serum samples from 26 HCW who recovered from mild forms of SARS-CoV-2 infection. All these HCW displayed seroconversion at D21 after symptom onset, and elicited an NAb response to SARS-CoV-2 correlated with the anti-RBD antibody levels. However, this neutralizing activity appears to be transient, with NAbs decreasing in titer, or even being completely lost, in association with a decrease in systemic IgA antibody levels from 2 months after disease onset.

Our findings are consistent with a very high rate of seroconversion and a temporal pattern of antibody appearance similar to that reported for hospitalized SARS-CoV-2-infected patients in other studies[6–8,10,14,21,22]. It has been reported for SARS-CoV that the peak level of IgG against whole-virus lysate is reached between 4 and 6 months after symptom onset, and that these levels then remain stable for at least 1 year before decreasing[16,17]. Here, for SARS-CoV-2, IgG antibodies are detected earlier, for anti-N and anti-S antibodies, and their levels seem to remain stable between M2 and M3.

We studied the neutralizing activity of the serum samples from these HCW, a biological marker often correlated with protective immunity[19], using a sensitive and reproducible virus neutralization test. The VNT determines the functional ability of antibodies to prevent virus infection in vitro. This assay has also the advantage to study the overall serum neutralizing activity (both isotypes and epitope-specific antibodies), as it is performed on a whole replicating virus. All the serum samples studied had neutralizing activity against a local clinical isolate of SARS-CoV-2 at D21, indicative of a potent protective immune response to SARS-CoV-2 infection, even after mild forms of COVID-19.

Antibodies directed against the S protein are thought to be responsible for neutralizing activity against this virus, particularly those recognizing the RBD epitope, because S protein has been shown to be essential for viral particle entry into target cells, via interactions between the RBD and angiotensin-converting enzyme 2 (ACE2). Moreover, S protein was identified from the structural proteins of SARS-CoV as a major inducer of protective immunity[23,24]. This suggests that it may be important to assess changes in the levels of antibodies directed against S protein and RBD over time, to improve our understanding of the potential protective immune response to SARS-CoV-2 infection. In our

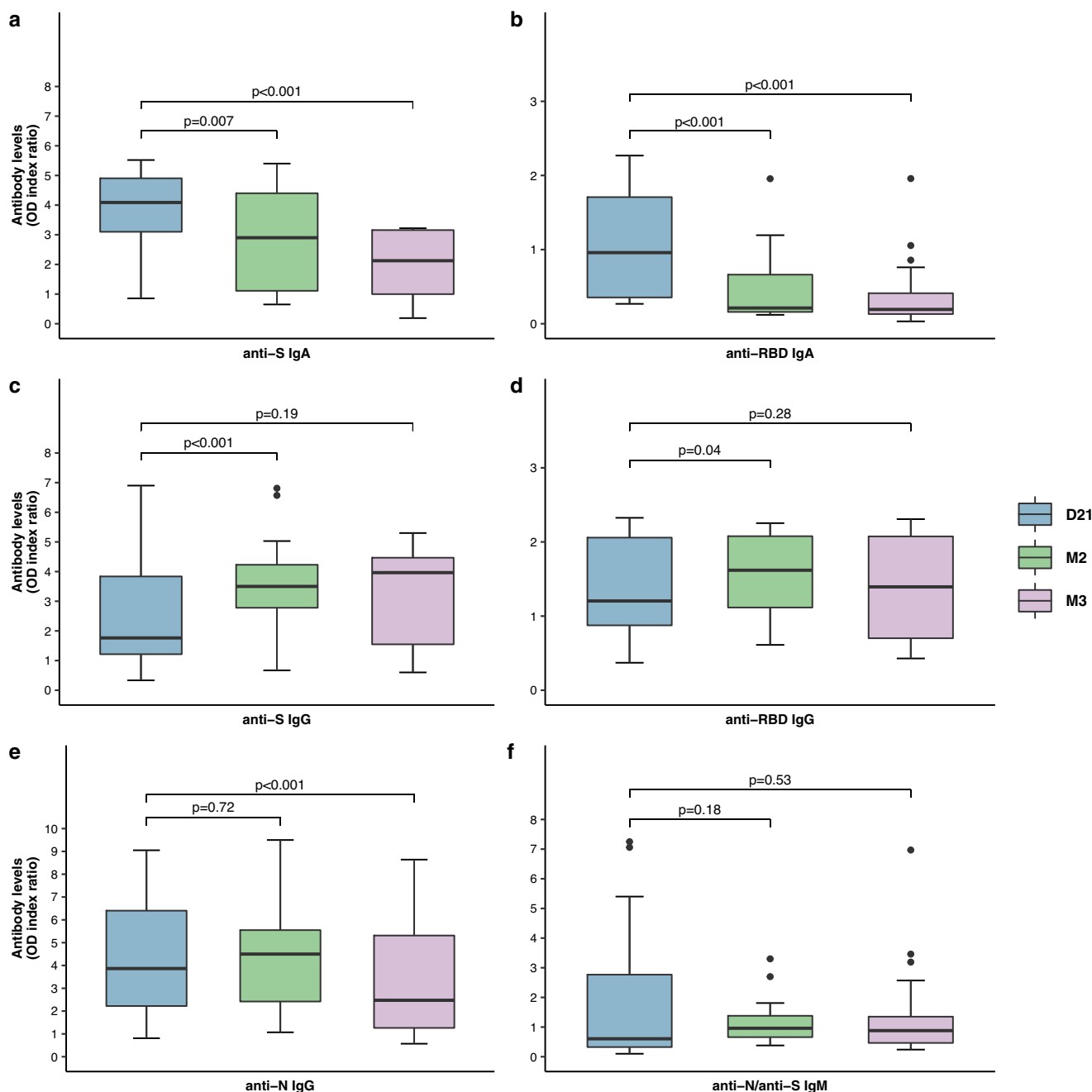

**Fig. 1 Changes in antibody levels from 3 weeks (D21) to 2 months (M2) and 3 months (M3) after severe acute respiratory syndrome coronavirus 2 (SARS-CoV-2) infection.** Sequential analysis of isotypic antibody responses (IgG, IgA, and IgM antibodies) to various SARS-CoV-2 viral components (N nucleocapsid protein, S spike protein, RBD receptor-binding domain) for serum samples collected from 26 HCW at three time-points: D21 ($n = 26$), M2 ($n = 17$), and M3 ($n = 26$) after symptom onset. Serum samples were tested in six commercial immunoassays, the positive cut-off values of optical density index ratio (OD index ratio) were 0.8 for anti-N IgG, 1.1 for anti-S IgG or IgA and anti-N/anti-S IgM, and 5× the cutoff value for anti-RBD IgG or IgA. The y-axes of the various panels are drawn to different scales. The D21 point is shown in blue, the M2 point in green, and the M3 point in pink. Horizontal lines indicate median values; boxes indicate quartiles 1 and 3; whiskers indicate the 1.5 interquartile range; black circles indicate outliers. *P* values were determined in two-sided Wilcoxon signed-rank tests. Source data are provided as a Source Data file.

experiments, IgA antibodies were the only antibody isotype to display rapid decreases in both levels and seropositivity rates from M2 onwards. Moreover, anti-RBD IgA antibodies, together with anti-RBD IgG antibodies, displayed the strongest association with NAb titers. This isotype also dominated the early neutralizing activity of the sera, as shown by neutralization experiments on the purified IgA fraction. These findings suggest that IgA antibodies are the major component of the NAbs developed in response to SARS-CoV-2 infection.

Nevertheless, the increase in IgA $EC_{50}$ at M3 may be explained by the proportion of specific IgA directed against SARS-CoV-2 RBD relative to total systemic IgA decreasing sharply, whereas SARS-CoV-2 anti-RBD IgG levels increased, potentially with an increase in neutralizing activity by somatic hypermutation. However, this increase in both the level and neutralizing activity of IgG directed against S protein may not be sufficient to compensate entirely for the disappearance of neutralizing activity due to IgA. This conclusion is also supported by the presence of anti-S

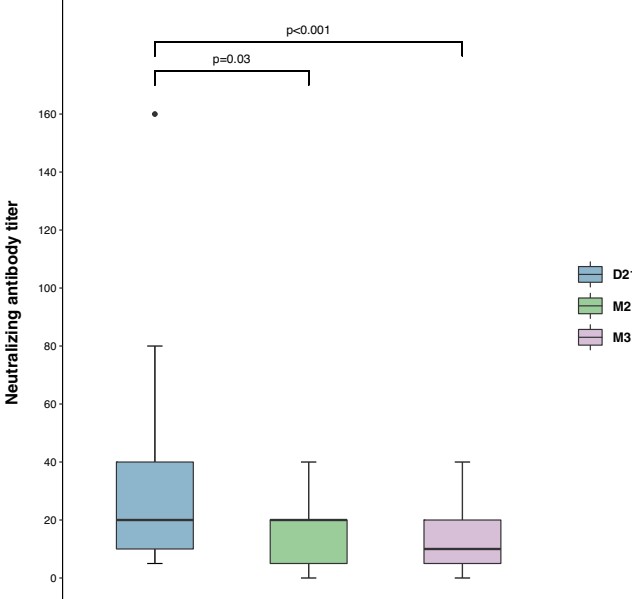

**Fig. 2 Dynamic changes in neutralizing antibody (NAb) titers between 21 days (D21), 2 months (M2), and 3 months (M3) after symptom onset.** Sequential analysis of NAb titers for serum samples from 26 HCW at three time points (D21, M2, and M3) after severe acute respiratory syndrome 2 (SARS-CoV-2) infection. The NAb titers of the various serum samples were measured with a whole-virus replication assay and are expressed as the highest serum dilution yielding 100% inhibition of the cytopathic effect. Number of participants: D21 ($n = 26$), M2 ($n = 17$), and M3 ($n = 26$). The D21 point is shown in blue, the M2 point in green, and the M3 point in pink. Horizontal lines indicate median values; boxes indicate quartiles 1 and 3; whiskers indicate the 1.5 interquartile range; black circles indicate outliers. $P$ values were determined in two-sided Wilcoxon signed-rank tests. Source data are provided as a Source Data file.

IgA in HCW who were seronegative for anti-S IgG at D21 and the loss of neutralizing activity for serum samples testing positive for anti-S or anti-RBD IgG, but negative for anti-S or anti-RBD IgA. These findings may be explained by the natural course and immunogenicity of SARS-CoV-2 infection, with a probable preferential induction of local and systemic, but not very long-lasting IgA antibodies against the S protein due to the mucosal entry pathway of this respiratory virus.

A recent study in rhesus macaques experimentally infected with SARS-CoV-2 showed the development of NAbs associated with protection against re-infection after a challenge at 35 days[25]. This result, consistent with our findings and those of other groups, indicates that SARS-CoV-2 infection elicits a protective humoral response, at least temporarily, which may be correlated with the neutralizing activity of sera. Further studies are required to assess the protection correlates after SARS-CoV-2 infection, such as the minimal NAb titer for protection, the contribution of secretory IgA and local immunity, and the neutralizing capacity of anti-S IgG over time, even after the disappearance of circulating IgA.

In conclusion, our results and the recent description of rare cases of SARS-CoV-2 reinfection[26,27] are in favor of a potent risk of re-infection in HCW and support strong recommendations to maintain an adequate use of personal protective equipment, and to apply infection prevention and control measures, even for HCW who have recovered from a mild form of COVID-19 and who have antibodies detectable in routine serologic tests, at least until strong correlates of immune protection are identified. This study also suggests that vaccination against SARS-CoV-2 in humans will probably have short lasting protective effect meaning that most people should get vaccinated periodically.

## Methods

**Study population and sampling.** This study was conducted at Pitié-Salpêtrière University Hospital, one of the first hospitals in Paris to manage patients infected with SARS-CoV-2. Symptomatic HCW with a SARS-CoV-2-positive RT-PCR on a nasopharyngeal specimen were included, and serum samples were obtained

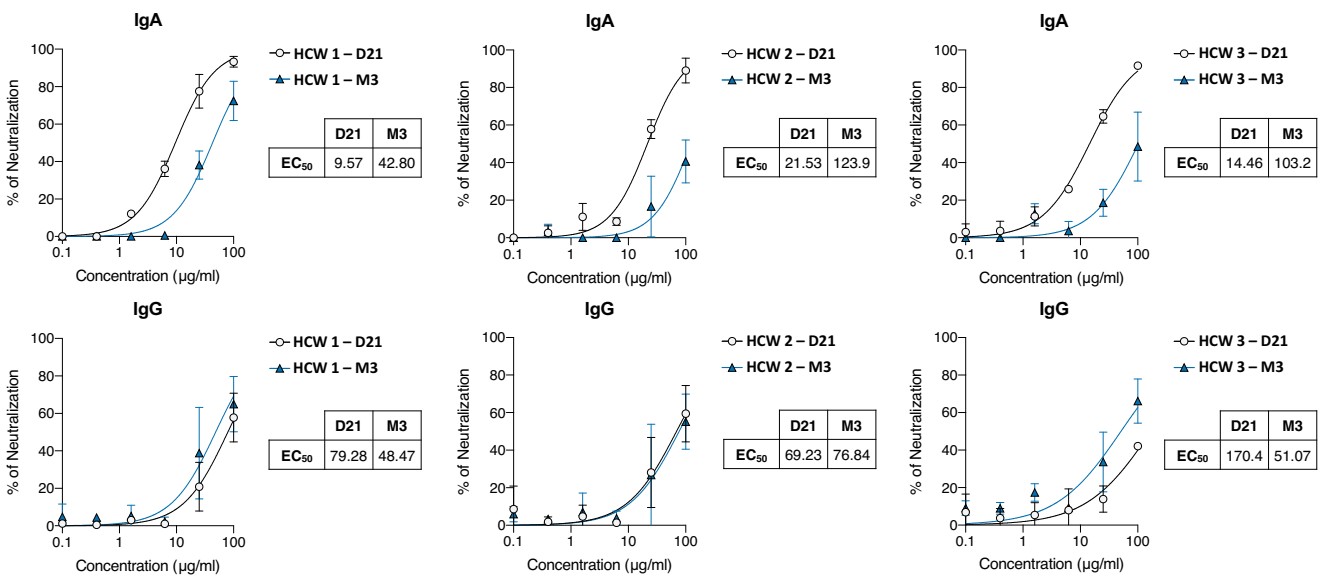

**Fig. 3 Neutralizing activity of purified IgA and IgG antibodies against SARS-CoV-2 from the sera of three healthcare workers (HCW) at 21 days (D21) and 3 months (M3) after symptom onset.** For each pair of sera the percentage neutralization was calculated at different antibody concentrations in two different neutralization experiments based on syncytium formation with S-Fuse cells. The half-maximal effective concentration ($EC_{50}$) values (µg/ml) were calculated and represent the neutralizing activity of each isotype. Data are presented as mean values ± standard deviation (SD). Non-linear regression curves were drawn. The black curves represent experiments performed on samples from D21 and the blue curves represent experiments performed on samples from M3. Source data are provided as a Source Data file.

between 2 and 4 weeks (D21), and then 2 months (M2) and 3 months (M3) after symptom onset. The study was carried out in accordance with the Declaration of Helsinki. This study was a retrospective non-interventional study with no addition to standard care procedures. Reclassifications of biological remnants into research material after completion of the ordered virological tests and data collection were declared to the Sorbonne Université Data Protection Committee under number 2020-025 in accordance with French law. Written informed consent for participation in this study was obtained from all participants.

**SARS-CoV-2 IgG, IgA, and IgM immunoassays**. Serum IgG, IgA, and IgM levels were determined with commercially available immunoassay kits: an automated chemiluminescent microparticle immunoassay for detecting IgG against SARS-CoV-2 nucleocapsid (N) protein (Architect SARS-CoV-2 IgG assay; Abbott Laboratories), two enzyme-linked immunosorbent assays (ELISA) for detecting IgG or IgA isotype antibodies against the S1 domain of the spike (S) protein (SARS-CoV-2 IgG or IgA ELISA, EuroImmun), two ELISA for detecting IgG or IgA isotype antibodies against the receptor-binding domain (RBD) of the S protein (Anti-SARS-CoV-2 IgG or IgA ELISA, Mediagnost), and one ELISA for detecting IgM against SARS-CoV-2 N and S proteins (Dia.Pro COVID-19 IgM, Dia.Pro Diagnostic Bioprobes srl). All immunoassays were used and interpreted according to the manufacturer's instructions. Samples for which the results were unclear were retested and considered positive if confirmed to be equivocal, and negative otherwise.

**Virus neutralization test**. The neutralizing activity of the various serum samples was assessed with a whole virus replication assay using a SARS-CoV-2 strain (GenBank accession number MW322968) isolated from a COVID-19 patient hospitalized at Pitié-Salpêtrière University Hospital. The infection of this patient was confirmed by a positive SARS-CoV-2 RT-PCR on 6 March 2020, and, the virus was isolated by inoculating Vero cells with a sputum sample in our biosafety level-3 (BSL-3) facility. The serum samples were decomplemented by heat inactivation (56 °C for 30 min), subjected to serial twofold dilution (1:5 to 1:2560), and incubated with 50 μl of diluted virus ($2 \times 10^3$ $TCID_{50}$/ml) in a 96-well plate at 37 °C for 60 min. We then added 100 μl of a Vero E6 cell suspension ($3 \times 10^5$ cells/ml) to the mixture and incubated at 37 °C under an atmosphere containing 5% $CO_2$ until the microscopy examination on day 4 to assess the cytopathic effect (CPE). Neutralization antibody titers are expressed as the highest serum dilution displaying 100% inhibition of the CPE. The same positive serum was added to each experiment to assess the reproductivity and specificity of the VNT assay; pre-epidemic serum samples from patients with other HCoV infections (confirmed by RT-PCR with FilmArray® Respiratory Panel 2 plus, Biomérieux) were also tested.

**Purification of IgA and IgG antibodies from serum**. Serum samples were diluted 1:3 in 1× PBS and incubated for 20 min with Protein G/agarose (Invivogen). IgG-depleted serum was then incubated with peptide M/agarose (Invivogen) for 20 min. Agarose beads were washed three times with 1 ml 1× PBS and incubated with 0.1 M glycine pH 2 for 10 min in ice. The pH was adjusted to 7 with 1 M Tris pH 9, and the eluted IgG and IgA were subjected to buffer exchange in 1× PBS (Vivaspin 500, Sartorius), through a 50 kDa membrane. The concentrations of IgA and IgG were determined by ELISA (Bethyl Laboratories), according to the manufacturer's instructions.

**Neutralization experiments in S-Fuse cells**. U2OS-ACE2 GFP1-10 or GFP 11 cells, also known as S-Fuse cells, become GFP-positive when they are productively infected by SARS-CoV-2 (ref. [20]). Cells were mixed (ratio 1:1) and plated at a density of $8 \times 10^3$ per well in a μClear 96-well plate (Greiner Bio-One). SARS-CoV-2 (BetaCoV/France/IDF0372/2020) (MOI 0.1) was incubated with IgA or IgG at the indicated concentrations for 30 min at room temperature and added to S-Fuse cells. The cells were fixed, 18 h later, in 2% paraformaldehyde, washed and stained with Hoechst stain (dilution 1:1000; Invitrogen). Images were acquired with an Opera Phenix high-content confocal microscope (Perkin Elmer). The area displaying GFP expression and the number of nuclei were quantified with Harmony software 4.8 (Perkin Elmer). The percentage neutralization was calculated from the GFP-positive area as follows: 100 × (1 − (value with IgA/IgG − value in "non-infected")/(value in "no IgA/IgG" − value in "non-infected")). The neutralizing activity of each isotype is expressed as the half maximal effective concentration ($EC_{50}$). We calculated $EC_{50}$ values (μg/ml) from a reconstructed curve of the percentage neutralization at the various concentrations indicated.

**Statistical methods**. Continuous variables are expressed as the median and IQR and discrete variables are expressed as numbers and percentages. Statistical analysis was performed with non-parametric tests in GraphPad Prism 6.0. A probability value of $p < 0.05$ was considered statistically significant. Spearman's rank correlation coefficients ($r_s$) were estimated to determine the associations between serum neutralization titers and Ig levels. The results were interpreted according to the strength and direction ($r$) and the statistical significance ($p$ value) of the correlation. Two-sided Wilcoxon signed-rank tests were used to compare serum Ig levels and NAb titers for HCW sera between D21 and M2 and between D21 and M3.

**Reporting summary**. Further information on research design is available in the Nature Research Reporting Summary linked to this article.

## Data availability
The GenBank accession code for the SARS-CoV-2 strain used in the VNT is MW322968. Source data are provided with this paper.

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

## Acknowledgements
The AGM laboratory is funded by Agence Nationale de Recherches sur le SIDA et les hépatites virales (AC43, Medical Virology), the SARS-CoV-2 Program of the Faculty of Medicine of Sorbonne University, the Agence Nationale de la Recherche. The GG laboratory is supported by the Fondation de France, "Tous unis contre le virus" framework Alliance (Fondation de France, AP-HP, Institut Pasteur) in collaboration with the Agence Nationale de la Recherche (ANR Flash COVID19 program) and by the SARS-CoV-2 Program of the Faculty of Medicine of Sorbonne University ICOViD programs. The OS laboratory is funded by Institut Pasteur, the ANRS, the Vaccine Research Institute (ANR-10-LABX-77), Labex IBEID (ANR-10-LABX-62-IBEID), "TIMTAMDEN" ANR-14-CE14-0029, "CHIKV-Viro-Immuno" ANR-14-CE14-0015-01, the Gilead HIV cure program, ANR/FRM Flash Covid PROTEO-SARS-CoV-2 and IDISCOVR, and Fondation pour la Recherche Médicale. We thank Nathalie Aulner and the UtechS Photonic BioImaging (UPBI) core facility (Institut Pasteur), a member of the France BioImaging network, for image acquisition and analysis.

## Author contributions
A.-G.M., V.C., S.B., D.B., D.S., O.S. and G.G. planned the research; S.M., I.M., V.L., K.Z., A.G., K.D. and T.B. conducted the experiments; V.L., S.M., I.M., D.P., A.J. and T.B. analyzed the data; S.M., A.-G.M. and V.C. wrote the paper. All the authors read and corrected the manuscript and approved the final version.

## Competing interests
The authors declare no competing interests.

## Additional information

## the Sorbonne Université SARS-CoV-2 Neutralizing Antibodies Study Group

Valérie Attali[5], Isabelle Baresse[5], Alexandra Beurton[5], Jacques Boddaert[5], Julie Bourmaleau[5], Martin Catala[5], Alexandre Demoule[5], Violaine Dunoyer[5], Cristina Esteban-Amarilla[5], Pierre Hausfater[5], Noémie Haziot[5], Queyras Ip[5], Nathalie Kubis[5], Laurence Lhoest[5], Catherine Lubetzki[5], Fabienne Marion[5], Elise Morawiec[5], Leila Mourtada[5], Brigitte Orcel[5], Capucine Morelot-Panzini[5], Mathieu Raux[5], Christophe Reinhard[5], Claire Riquier[5], Xavier Roubertier[5], Nicolas Weiss[5] & Bernard Zalc[5]

[5]Sorbonne Université, Assistance Publique-Hôpitaux de Paris, Pitié Salpêtrière Hospital, Paris, France.

