## [Peer Review File · Nature Communications]

Reviewer comments, first round:

Reviewer #1 (Remarks to the Author):

. I have carefully read this manuscript. Marot ¹¹¹_{ISEP} et al studied antibody levels, especially anti-spike and neutralizing antibodies, at D15 and M2 for 16 healthcare workers. Overall, due to the small patient number and only two samples for each patient. It is very difficult to draw any reliable conclusions. This study provides little useful information for the field.

There are some major concerns in this study:

1. The number of patient is 16. It has been reported that serum antibody levels are affected age, disease severity and other factors. There is a great individual diversity in antibody responses. May not be able to draw a conclusion from such a small sample size.
2. Spike protein, or S1 region is a large protein in size, which may have many neutralizing and non-neutralizing epitopes. If use RBD-based kits, the antibody levels are more related to neutralization.

As a result, I suggest rejection of this manuscript.

Reviewer #2 (Remarks to the Author):

The study by Marot and colleagues discusses how anti-SARS-CoV-2 serum IgG, IGA and IgM binding antibodies and serum neutralizing antibody responses evolve during a two month period from the onset of clinical symptoms, in a small number of health care workers that did not require hospitalization during COVID-19 infection. The authors report that during the period of observation, the anti-S IgG antibody levels increased, while the anti-S IgA antibody levels decreased. Interestingly, they report that the serum neutralizing activities also decreased (and in some cases became undetectable) during this short period of observation.

Although the observation that during this rather short period of time, the serum neutralizing activities decreased, possible underlying mechanisms for this decline are not discussed. Based on the observed changes in IgG and IgA levels during the period of observation, one could hypothesize that the IgA, but not IgG, component is responsible for the serum neutralizing activities. That would be an important point to prove experimentally. The authors could determine the relative importance of the IgG fraction in serum neutralizing activities, by adsorbing the IgG fraction from sera at 15 days and at two months. Similar experiment could be performed with the IgA fraction.

In the Abstract (lines 30-31) the authors state that" 'neutralizing antibodies decreased at M2.....in correlation with anti-S IgA levels decrease' but on line 105 they state: ' Nab titers were correlated with anti-S and IgA antibodies'. Could the authors clarify?

Also, they mention that the titers of neutralizing antibodies correlated with anti-N IgG antibody levels. As antibodies to N are not expected to neutralize the virus, this observation needs discussion. In fact, I don't see how the anti-N IgG antibody levels correlated with serum neutralizing titers as the anti-N IgG levels remained unchanged over the period of observation.

A key element that is missing is the absence of information on the viral loads in those patients. Where these patients viremic? Did their viral loads decrease over the observation period? i.e., as the neutralizing antibody responses decreased?

Also, is there any information whether any of these subjects were re-infected once their serum neutralizing antibody responses decreased?

Reviewer #3 (Remarks to the Author):

The present study by Marot et al described the neutralizing antibodies against SARS-CoV-2 among infected healthcare workers (HCW). They compared the NAb, IgG, IgA and IgM levels at day 15 and month 2 PSO and found that NAb as well as immunoglobulins decreased at M2. They concluded that SARS-CoV-2 induced short-lasting humoral immune protection in HCW. Currently, the dynamics of NAb in COVID-19 have been reported by several groups (Yao et al. medRxiv preprint doi: <https://doi.org/10.1101/2020.07.18.20156810>; Wang et al. J Clin Invest. 2020. <https://doi.org/10.1172/JCI138759>; Juno et al. Nat. Med. 2020 Jul 13; DOI : 10.1038/s41591-020-0995-0). The experiment designs are incomplete and provided limited information.

Major points:

1. In order to describe the dynamics of neutralizing antibodies and humoral response in HCW infected with COVID-19, there should be more than two time-point examination. Moreover, due to the small samples sizes (only 16 HCW with COVID-19 mild-forms), there is a concern regarding the interpretation of these results.
2. Why the anti-S IgM titers increased at M2 when compared with that at day 15? IgM is the antibody isotype to appear early during virus infection. The authors should give some discussion.

Minor points:

1. Lack of Table describing the patient characteristics.
2. The important data, such as neutralizing capacity of other coronaviruses and the correlation analysis between Nab titers and Ig antibody levels, are missing.

Institut Pierre Louis d'Épidémiologie et de Santé Publique
Pierre Louis Institute of Epidemiology and Public Health

Unité mixte de recherche en santé n° 1136 (UMR-S 1136)
Directrice : Dominique Costagliola

Editorial Office

Paris, 22th October 2020

Reviewer #1

Overall, due to the small patient number and only two samples for each patient. It is very difficult to draw any reliable conclusions. This study provides little useful information for the field. There are some major concerns in this study:

1. The number of patients is 16. It has been reported that serum antibody levels are affected age, disease severity and other factors. There is a great individual diversity in antibody responses. May not be able to draw a conclusion from such a small sample size.

We fully agree with this comment and in order to consolidate our results we have recruited 10 new patients. Our study now covers 26 patients studied at D21 and M3, 17 of them also having an M2 point.

2. Spike protein, or S1 region is a large protein in size, which may have many neutralizing and non-neutralizing epitopes. If use RBD-based kits, the antibody levels are more related to neutralization.

In order to take your remark into account, we have quantified the IgA and IgG antibodies specifically directed against the SARS-CoV-2 RBD using the ELISA Anti-SARS-CoV-2 IgA or IgG kits from Mediatech (results page 5, lines 99 to 118 and figure 1).

Reviewer #2

Although the observation that during this rather short period of time, the serum neutralizing activities decreased, possible underlying mechanisms for this decline are not discussed. Based on the observed changes in IgG and IgA levels during the period of observation, one could hypothesize that the IgA, but not IgG, component is responsible for the serum neutralizing activities. That would be an important point to prove experimentally. The authors could determine the relative importance of the IgG fraction in serum neutralizing activities, by adsorbing the IgG fraction from sera at 15 days and at two months. Similar experiment could be performed with the IgA fraction.

As suggested by the reviewer, we have demonstrated the relative implications of IgA and IgG in the serum neutralizing activities. For this, we have purified and quantified the IgA and IgG antibodies in sera from D21 and M3 for 3 patients, according to the procedure described in the material and methods section on page 11 (lines 256 to 263). Next, we have performed additional neutralization experiments using syncytia formation in a fractional GFP system to calculate half-maximal effective concentration (EC50) for each isotype (technique described in material and methods pages 11-12 (lines 264 to 278). The results confirm that the IgA antibodies exhibit a higher efficiency to neutralize the SARS-CoV-2 in the early antibody response than the IgG antibodies. The decrease in the neutralizing activity of IgA (i.e EC50) could be explained by the decrease of SARS-CoV-2 specific-IgA proportion in sera and be correlated with the decrease in the serum neutralizing activities over time (results page 7, lines 145 to 150 and figure 3 and discussion page 8, lines 188 to 196).

In the Abstract (lines 30-31) the authors state that” ‘neutralizing antibodies decreased at M2.....in correlation with anti-S IgA levels decrease’ but on line 105 they state: ‘ Nab titers were correlated with anti-S IgG and IgA antibodies’. Could the authors clarify?

We agree with the reviewer that the paragraph "NAb titers and Ig antibody levels" on line 116 is confusing. For this, we have modified the paragraph with the new data (in D21, M2 and M3) and expressed it differently. "Association of the NAb titer with the IgG, IgA and IgM antibody profile"(results page 6, lines 139 to 144). Indeed, these results reflect indirectly that the overall level of IgA and IgG antibodies, quantified in the sera of the subjects studied, is correlated with the neutralization titers. In other words, the more antibodies there are in the serum (neutralizing and non-neutralizing), the more the serum neutralizes.

Also, they mention that the titers of neutralizing antibodies correlated with anti-N IgG antibody levels. As antibodies to N are not expected to neutralize the virus, this observation needs discussion. In fact, I don't see how the anti-N IgG antibody levels correlated with serum neutralizing titers as the anti-N IgG levels remained unchanged over the period of observation.

We agree with this comment that this point needs discussion. Actually, correlation between anti-N IgG antibody levels and NAb titers is probably explained by the fact that the levels of SARS-CoV-2 specific antibodies is a reflect of the efficiency of the immune response. As it was previously discussed, the neutralizing activity of sera is correlated with the anti-RBD antibody levels (mostly anti-RBD) and it is thus expected that high level of anti-S antibodies is associated with high levels of anti-N antibodies. So, we have suppressed this part because anti-N IgG is one of the less correlated isotype with NAb titers (as reported by the lowest spearman's rank correlation coefficient) and to avoid any confusion, we have only presented isotypes with spearman's rank correlation coefficient > 0.6 (i.e anti-RBD IgA and IgG) and p value < 0.05 (results page 6, lines 139 to 144).

A key element that is missing is the absence of information on the viral loads in those patients. Where these patients viremic? Did their viral loads decrease over the observation period? i.e., as the neutralizing antibody responses decreased?

As requested by the reviewer, we have added information about the initial viral loads at the time of the diagnostic (estimated by the cycle threshold of the RT-PCR) in the Table 1. Initially, there was no follow-up on the RT-PCR results in the nasopharyngeal sphere because these HCW were placed in isolation at home for at least 7 days. For a few of them, other RT-PCR were performed a few weeks or months after the COVID-19 episode and all the RT-PCR results were negative. Unfortunately, we do not have information about SARS-CoV-2 RNAemia as it was not recommended, and no monitoring of SARS-CoV-2 RNA was performed in blood (i.e plasma).

In addition, it has been described in some studies (Hagman & al., Clin. Infect Dis. DOI: 10.1093/cid/ciaa1285 ; Eberhardt & al., Viruses DOI: 10.3390/v12091045 ; Tan & al., Braz J Infect Dis. DOI: 10.1016/j.bjid.2020.08.010) that RNAemia could be a predictor or a marker of severe outcome. As, all the HCW followed in the present study had mild to moderate COVID-19, we can assume that none of them were viremic and we have not focused on this parameter.

Also, is there any information whether any of these subjects were re-infected once their serum neutralizing antibody responses decreased?

Until the point M3, as far as we know none of the subjects have been reinfected. HCW are now particularly attentive to barrier gestures and during this summer COVID-19 prevalence in France was at a lower rate than now. As part of this work, we have set up a follow-up of all the subjects enrolled in this study with regular time-points and potential reinfection is carefully under surveillance.

Reviewer #3

Currently, the dynamics of NAbs in COVID-19 have been reported by several groups (Yao et al. medRxiv preprint doi: <https://doi.org/10.1101/2020.07.18.20156810>; Wang et al. J Clin Invest. 2020. <https://doi.org/10.1172/JCI138759>; Juno et al. Nat. Med. 2020 Jul 13; DOI : 10.1038/s41591-020-0995-0). The experiment designs are incomplete and provided limited information.

Major points:

1. In order to describe the dynamics of neutralizing antibodies and humoral response in HCW infected with COVID-19, there should be more than two time-point examination. Moreover, due to the small samples sizes (only 16 HCW with COVID-19 mild-forms), there is a concern regarding the interpretation of these results.

As responded to the first referee, we have added 10 new subjects in the study to consolidate our results. Also, we have now a new point at M3 for all the subjects, and 17 of them also have a M2 point.

2. Why the anti-S IgM titers increased at M2 when compared with that at day 15? IgM is the antibody isotype to appear early during virus infection. The authors should give some discussion.

It is true that, as the reviewer points out, the level titer of anti-S IgM increases slightly, although not significantly, between D21 and M2. We then note a slight decrease at M3 (new point tested) still not significant. Given the fact the values are rather low, we believe that these results rather reflect the limits of the test used and that in any case, this test shows that the IgM level does not vary significantly and could question place of IgM antibodies in the COVID-19 diagnosis.

Minor points:

1. Lack of Table describing the patient characteristics.

As requested by the reviewer, we have added a Table including all the patient's characteristics (Table 1).

2. The important data, such as neutralizing capacity of other coronaviruses and the correlation analysis between Nab titers and Ig antibody levels, are missing.

To investigate cross-neutralization activities between Low-Pathogenic Human Coronaviruses (LP-HCoV ; i.e HCoV OC43, 229E, NL63 and HKU1) we have performed our VNT on 10 patients sera with RT-PCR confirmed LP-HCoV infections and we did not find any neutralizing activities for these sera at a NAb titers $\geq 1/5e$ (page 6, lines 136 to 138). Then, in order to assess potential serology cross-reactivities, these LP-HCoV sera were also tested in all ELISA experiments and no positivity results were found for any of them. From our experiments we did not observe cross-reactivities or neutralization between LP-HCoV and SARS-CoV-2 (results page 6, lines 121 to 125 and lines 136 to 138). Moreover, SARS-CoV-2 is most closely related to SARS-CoV (Betacoronavirus, clade B), so we can expect that probability of cross-neutralization is most likely between the SARS-CoV-2 and the SARS-CoV. However, Anderson & al., Emerging Microbes & Infections DOI : 10.1080/22221751.2020.1761267 have shown that sera from SARS-CoV recovered patients (still neutralizing even 17 years after recovery) do not cross-neutralize with the SARS-CoV-2. Similar results were found by Yang & al., EbioMedicine DOI : 10.1016/j.ebiom.2020.102890

Sincerely,

Dr Stéphane MAROT, M.D., M.Sc.

Reviewer comments, second round:

Reviewer #1 (Remarks to the Author):

Dear authors:

Thanks for revising. All of my concerns are properly addressed.

Reviewer #2 (Remarks to the Author):

The authors recruited additional patients and performed additional experiments that support their overall conclusions regarding the kinetics of binding and neutralizing antibody responses during SARS-CoV-2 infection. I do not have any major comments.

Minor points:

Line 91: fix the reference error

Lines 100-112: indicate which the figure(s) are associated with the discussed data

Figure 1: Indicate in the legend that the Y axes of the different panels are in different scale. For each panel indicate which data points are from D21, M2 and M3

Figure 2: The y axis can be reduced as the neutralizing antibody titers are >80re missing.

Institut Pierre Louis d'Épidémiologie et de Santé Publique
Pierre Louis Institute of Epidemiology and Public Health

Unité mixte de recherche en santé n° 1136 (UMR-S 1136)
Directrice : Dominique Costagliola

Editorial Office

Paris, 3rd December 2020

Reviewer #1

Dear authors:

Thanks for revising. All of my concerns are properly addressed.

Thanks for your interesting remarks during the first reviewing which allowed us to improve our manuscript.

Reviewer #2

The authors recruited additional patients and performed additional experiments that support their overall conclusions regarding the kinetics of binding and neutralizing antibody responses during SARS-CoV-2 infection. I do not have any major comments.

Minor points:

Line 91: fix the reference error

The reference has been fixed.

Lines 100-112: indicate which the figure(s) are associated with the discussed data

This paragraph is about the rate of seropositivity of the different serological tests performed. We only express the results in the text and do not provide a figure for these results. However, data used to calculate seropositivity rate are provided in the Source Data file with this paper.

Figure 1: Indicate in the legend that the Y axes of the different panels are in different scale. For each panel indicate which data points are from D21, M2 and M3

We have added this information in the legend of the figure 1. Also, the color used for each panels are present in the right of the panels and have been explicited in the legend.

Figure 2: The y axis can be reduced as the neutralizing antibody titers are >80

We prefer to not reduce the Y axis in the purpose to show the outlier point at a titer of 160.

Sincerely,

Dr Stéphane MAROT, M.D., M.Sc.